# BiGym: A Demo-Driven Mobile Bi-Manual Manipulation Benchmark

**Nikita Chernyadev**[*]   **Nicholas Backshall**[*]   **Xiao Ma**[*]
**Yunfan Lu**   **Younggyo Seo**   **Stephen James**

Dyson Robot Learning Lab

**Abstract:** We introduce BiGym, a new benchmark and learning environment for mobile bi-manual demo-driven robotic manipulation. BiGym features 40 diverse tasks set in home environments, ranging from simple target reaching to complex kitchen cleaning. To capture the real-world performance accurately, we provide *human-collected* demonstrations for each task, reflecting the diverse modalities found in real-world robot trajectories. BiGym supports a variety of observations, including proprioceptive data and visual inputs such as RGB, and depth from 3 camera views. To validate the usability of BiGym, we thoroughly benchmark the state-of-the-art imitation learning algorithms and demo-driven reinforcement learning algorithms within the environment and discuss the future opportunities. Project website: https://chernyadev.github.io/bigym/

**Keywords:** Bi-Manual Manipulation, Mobile Manipulation, Benchmark

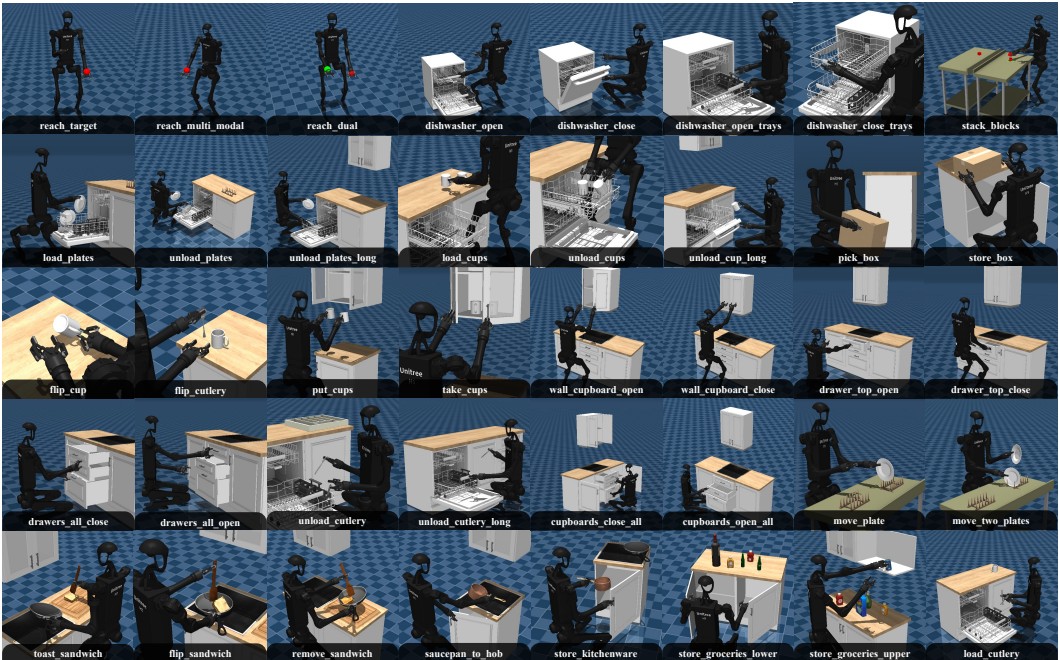

Figure 1: BiGym focuses on mobile manipulation with home assistance humanoids. We provide 40 tasks ranging from simple mobile target reaching to complex dishwasher manipulations. Specifically, each task comes with demonstrations recorded by human demonstrators and can be used to benchmark both imitation learning and reinforcement learning algorithms.

---

[*]equal contributions

8th Conference on Robot Learning (CoRL 2024), Munich, Germany.

# 1  Introduction

Machine learning benchmarks are of significant importance for measuring and understanding the progress of research algorithms. Examples of notable benchmarks include ImageNet [1] for image understanding, KITTI [2] for autonomous driving, and SQuAD for language-based question answering [3]. In robotics, prior benchmarks have greatly reduced the cost of iterating and developing algorithms. Examples include OpenAI Gym [4], DeepMind Control Suite [5], and MetaWorld [6]. However, all of these benchmarks focus on pure reinforcement learning (RL) with dense shaped rewards, limiting their application in long-horizon manipulation tasks where accurately defining reward functions is challenging.

While crafting reward is difficult, obtaining expert trajectories, such as those from human demonstrations, is relatively straightforward. This advantage has boosted the popularity of demonstration-driven methods within the robot learning community, manifesting as both imitation learning (IL) [7, 8, 9, 10, 11, 12, 13] and demo-driven RL [14, 15, 16, 17]. To support the research of building demo-driven agents, RLBench [18] was created with a wide range of single-arm fixed manipulation tasks with expert demonstrations generated by motion planners. Using motion planners allows RLBench to generate a large amount of demonstration data purely in simulation, however, the output trajectories are often either unnatural due to the inherent randomness in sampling-based planners, or have an unrealistically narrow trajectory distribution when compared to noisy real-world human demonstrations. Moreover, the community-progress is beginning to plateau on a large number of RLBench tasks, in particular with recent 3D next-best pose agents [10, 11, 16, 17, 19, 20, 21].

These limitation highlights the need for a new benchmark which provides: (1) more natural demonstrations like those seen in real-world robot data and (2) a set of new challenging tasks where state-of-the-art algorithms are likely to perform poorly. To this end, we present BiGym, a demo-driven mobile bi-manual manipulation benchmark with a humanoid embodiment. BiGym covers 40 visual mobile manipulation tasks, ranging from simple tasks like moving plates between drainers to interacting with articulated objects such as dishwashers (see Figure 1). Unlike prior humanoid benchmarks [22, 23] that focus only on RL with dense shaped reward functions, which may lead to undesired behaviors [24], we provide for each task only sparse rewards but with 50 demonstrations, allowing evaluation of both IL and RL algorithms. Additionally, compared to previous benchmarks that rely on expert demonstrations generated by planners [18], the human-collected demonstrations in BiGym are much more realistic and multi-modal (see Figure 3), better reflecting the trajectories of real-robot movements. Finally, BiGym considers locomotion and mobile bi-manual manipulation challenges separately; specifically, BiGym allows users to switch between the *whole-body* mode, which jointly considers locomotion and manipulation, and a *bi-manual* mode, which focuses on upper-body mobile manipulation while controlling the lower body with fixed controllers (see Figure 2). This separation of action modes enables researchers to better investigate and benchmark the capability of various algorithms with different focuses, i.e., locomotion control and mobile bi-manual manipulation solely. Code for BiGym is available on our project website.

# 2  Related Works

With the rapid progress in robot learning algorithms, the role of benchmarks has become crucial as a tool to understand the effect of various algorithmic design choices and compare algorithms in the same setup. There have been a series of benchmarks for complex manipulation tasks. Most of the existing benchmarks consider a single-arm manipulation scenario. The IKEA furniture assembly environment [25], BEHAVIOUR [26], and Habitat [27] provide a wide range of long-horizon household object (mobile) manipulation tasks. They emphasise long-horizon planning capabilities but employ abstract low-level actions that overlook physical interactions. Some benchmarks focus on more realistic settings with physics interactions [18, 28, 29, 30, 31, 32] and mainly support training RL agents. Notably, James et al. [18] provide APIs to generate expert demonstrations with motion planners. As a result, it is widely used for benchmarking IL [10, 11, 21] and demo-driven RL algorithms [16, 17, 19, 20]. Concurrent to our work, RoboCasa [33] constructs realistic environments

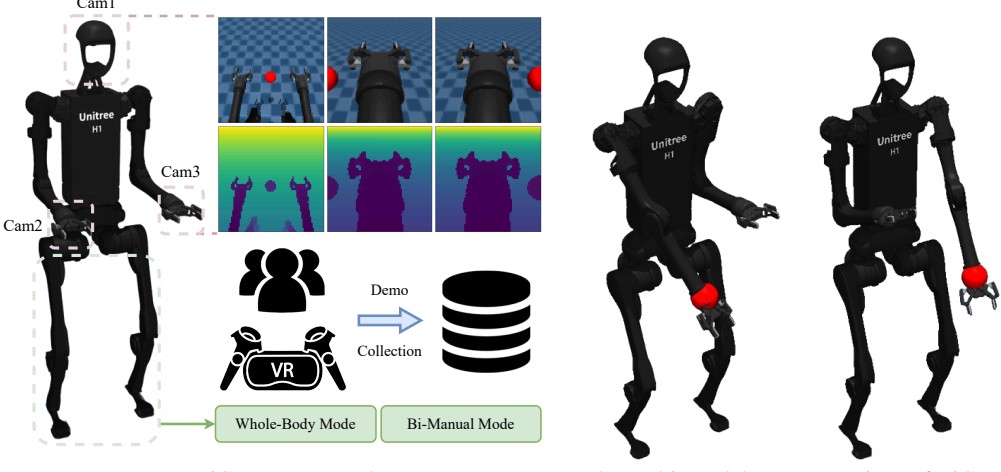

(a) BiGym Framework       (b) Multi-Modal Demonstrations of BiGym

Figure 2: (a) BiGym builds upon Unitree H1 robot with 3 RGB-D cameras at the head, left wrist, and right wrist. We collect human demonstrations by tele-operating with VR devices. BiGym allows users to control the humanoid in either **whole-body mode**, which considers both locomotion and manipulation, or the **bi-manual mode**, which simplifies the locomotion with a predefined controller for the lower-body. (b) BiGym provides human-collected multi-modal demonstrations for tasks, e.g., in `reach_target_multi_modal`, the agent can finish the task by reaching the target with either the left or right hand.

with human demonstrations, but only for single-arm tasks. Unlike these benchmarks that only consider a single-arm manipulation setup, BiGym provides a variety of mobile bi-manual manipulation tasks.

Bi-manual manipulation benchmarks consider controlling two arms or floating dexterous hands to interact with the environment [34, 35, 36]. More recently, benchmarks for humanoid robots have been introduced. For instance, LocoMujoco [22] focuses on locomotion control of different types of humanoids with two arms, but does not include manipulation tasks. As a concurrent work, HumanoidBench [23] focuses on benchmarking RL algorithms with task-specific shaped rewards on 15 locomotion and 12 manipulation tasks. In contrast, BiGym supports benchmarking both IL and RL algorithms by providing 40 tasks with human-collected demonstrations. These demonstrations exhibit realistic noisy trajectories that cover a wider data distribution compared to planner-generated demonstrations [18], thus enabling the evaluation that better reflects the real-world performance of algorithms. We provide a detailed comparison across benchmarks in Table 1.

## 3 BiGym

We present BiGym, a demo-driven mobile bi-manual manipulation benchmark. BiGym consists of 40 mobile bi-manual manipulation tasks, ranging from simple target reaching to complex dishwasher cleaning tasks. To evaluate IL and demo-driven RL algorithms in a realistic scenario with noisy, multi-modal demonstrations, BiGym provides human-collected demonstrations for all tasks. We describe which challenges BiGym presents (see Section 3.1), the simulation platform (see Section 3.2), and details on human demonstration datasets (see Section 3.3) and tasks (see Section 3.4).

### 3.1 Challenges of BiGym

We design BiGym to pose the following challenges:

**Partial Observability.** BiGym tasks are formulated as a partially observable Markov decision process (POMDP) [37] with discrete time $t = 1, 2, \ldots, T$, continuous action $a_t$, hybrid-observations, which include visual observations $o_t$ and robot low-level states $s_t$, and reward $r_t$. To achieve the task, the agent is required to learn a belief $b_t$, i.e., a distribution over the environment states, given past partial

| Benchmark | Mobile | # Arms | Action Mode | Task Horizon | Demonstrations | Human Demo | # Tasks |
|---|---|---|---|---|---|---|---|
| MetaWorld [6] | ✗ | 1 | J / EE | 500 | ✗ | ✗ | 50 |
| RLBench [18] | ✗ | 1 | J / EE | 100 - 1000 | ✓ | ✗ | 106 |
| RoboSuite [41] | ✓ | 1 / 2 | J / EE | 500 | ✓ | ✓ | 9 |
| LocoMujoco [22] | ✓ | 0* | J | 100 - 500 | ✗ | ✗ | 27 |
| HumanoidBench [23] | ✓ | 2 | J | 500 - 1000 | ✗ | ✗ | 27 |
| BiGym (ours) | ✓ | 2 | J / J + FB | 1000 - 7000 | ✓ | ✓ | 40 |

Table 1: Comparison with widely used benchmarks. **J**: Joint position action mode that controls all the joint angles of the robot. **EE**: End-Effector action mode with low-level planners. **FB**: Floating base.
*Although LocoMujoco considers humanoids, it only studies the locomotion tasks, rather than manipulation.

observations $\{\mathbf{o}_t, \mathbf{s}_t\}_{t=1}^T$ and actions $\{a_t\}_{t=1}^T$, which is non-trivial given the curse of history and the curse of dimensionality of POMDPs [38, 39].

**Complex Task Space.** Mobile bi-manual manipulation introduces a much more complex task space compared to fixed single-arm settings. This is because, with the presence of dual arms and the mobility of the agent, there may exist multiple ways of solving a single task. For example, to grasp a cup on the side of a table and put it into the closed drawer, the robot can consider several different ways: (1) pick up the cup with one hand, pull the drawer with the other hand, and put the cup into the drawer; (2) pull the drawer with one hand, pick up the cup with the same hand, and put the cup into the drawer. The mobility of the robot also allows navigating to the target with different routes. As a result, even if a mobile robot has a comparable number of degrees of freedom to a fixed robot, the task space is highly multi-modal and significantly more complex. To test the capabilities of robot learning algorithms in such scenarios, BiGym offers a wide range of tasks featuring complex task spaces and human-collected demonstrations with diverse modalities.

**Long Task-Horizon and Sparse Reward.** In common household scenarios, the agent will need to perform long-horizon tasks, which are composed of a series of sub-tasks, which require both task-level planning and low-level motion planning. For example, to load plates into the dishwasher, the agent should correctly locate the plates, open the dishwasher, pull out the trays, accurately put the plates on the trays, and finally, close the door. In addition, due to the complex task space, it is extremely difficult to properly define reward functions. If one can define such rewards, the agent may easily fall into local minimas by learning to exploit such sub-optimal shaped rewards. Thus, BiGym instead provides a set of sparsely-rewarded tasks along with noisy human-collected demonstrations, to evaluate the performance of IL and demo-driven RL algorithms in a more realistic setup.

**Realistic Multi-Modal Demonstrations.** In contrast to prior benchmarks that generate expert demonstrations with motion planners [18, 31, 40], BiGym provides human-collected demonstrations that are highly noisy and multi-modal. Specifically, we design BiGym tasks to be solvable in multiple ways to induce a multi-modal demonstration distribution. For instance, in `reach_target_multi_modal` task, reaching the target can be achieved with either left or right hand, as shown in Figure 2(b). This design enables us to evaluate the capabilities of robot learning algorithms using more realistic demonstrations, rather than synthetic demonstrations consisting of unnatural trajectories (see Figure 3).

### 3.2 Simulation Platform

We build BiGym simulation environments based on MuJoCo [42] (see Figure 2(a) for the illustration of the whole system). We brief the core design choices below and more details are in Appendix A.

**Humanoid Body Configurations.** We implement the platform with the Unitree H1 robot given its publicly available model [43]. As the original H1 comes with no grippers, we attach an additional Robotiq 2F-85 gripper with an actuated wrist joint to each arm. We note that it is easy to swap the parallel gripper with other dexterous manipulators, but we leave it for future study as we observe that parallel grippers are sufficient for current tasks.

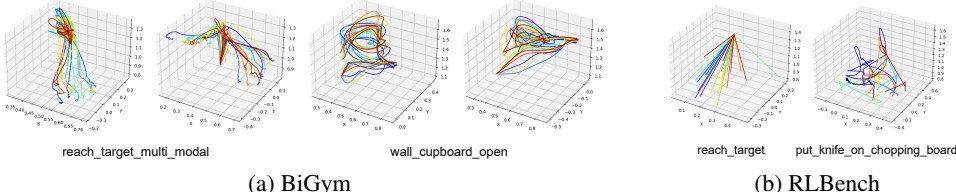

| (a) BiGym | (b) RLBench |

Figure 3: Visualisations of arm wrist position distributions of BiGym and RLBench. We visualise the wrist positions of both BiGym human collected trajectories on the `reach_target_multi_modal` and the `wall_cupboard_open` task, as well as the RLBench `reach_target` and the `put_knife_on_chopping_board` task. The trajectories of BiGym are noisy, multi-modal, but smooth in general, but the motion planner generated trajectories of RLBench are either straight lines or unnatural.

```
1    from bigym.envs.reach_target import ReachTarget
2    from bigym.action_modes import JointPositionActionMode
3    from demonstrations.demo_store import DemoStore
4    from demonstrations.utils import Metadata
5
6    env = ReachTarget(
7        action_mode=JointPositionActionMode(
8            floating_base=True, absolute=True,
9        )
10   )
11
12   demo_store = DemoStore.google_cloud()
13   demos = demo_store.get_demos(Metadata.from_env(env))
14
15   agent = Agent()
16   agent.ingest(demos)
17
18   obs, training_steps, episode_length = None, 100, ENV_TIME_LIMIT
19   for i in range(training_steps):
20       if i % episode_length == 0:
21           obs, _ = env.reset()
22       action = agent.act(obs)
23       obs, reward, terminated, truncated, info = env.step(action)
24       agent.add_to_buffer(obs, reward, terminated, truncated, info)
25       agent.update()
26   env.close()
```

Figure 4: Example usage of the BiGym Environment for training a reinforcement learning agent. Demonstrations are pulled from a remote store and cached locally. Users can also customise their action modes or use the off-the-shelf `JointPositionActionMode` with flags to switch between the *bi-manual* or *whole-body* action modes with either *absolute* or *delta* actions.

**Observation Spaces.** As shown in Figure 2(a), we mount three cameras on the robot: the forehead, the left wrist, and the right wrist. Each camera can generate both RGB and depth observations, which supports a diverse types of algorithms which use either type of observation. As a result, the observation space is defined as $\mathcal{O} = \{\mathcal{I}_{\text{head}}, \mathcal{I}_{\text{left}}, \mathcal{I}_{\text{right}}, \mathcal{D}_{\text{head}}, \mathcal{D}_{\text{left}}, \mathcal{D}_{\text{right}}, s_{\text{proprio}}\}$, where $\mathcal{I}$ is the RGB image, $\mathcal{D}$ is the depth image, and $s_{\text{proprio}}$ is the proprioception state of the robot. Additional observations, e.g., the gripper poses and robot poses, can also be easily obtained if required.

**Action Modes.** It remains unclear to the robotics community what action modes are the best for complex embodiment in mobile bi-manual manipulation tasks. Thus in BiGym, we provide flexible configurations for users to customise the action modes they want to use, and leave the choice to the users. Specifically, we provide two off-the-shelf action modes: the *whole-body* action mode and the *bi-manual* action mode, with either *delta* or *absolute* actions. For the whole-body action mode, we allow full control of the humanoid joints. This allows studying whole-body manipulation with locomotion. With the bi-manual action mode, we simplify the control by treating the lower-body of the humanoid as an omni-directional *floating base* controlled by classic controllers. In this case, we can focus on upper-body bi-manual mobile manipulation skills.

**Scenes.** The scenes in BiGym are created from MuJoCo MJCF models using a custom object-oriented API based on dm_control [5]. All MJCF models provided in BiGym were created from publicly available 3D models. Many other 3D models were processed to be used in BiGym: meshes were decimated to reduce the total number of polygons, moving parts of articulated objects were

separated, required joints and actuators were added, and convex collision meshes were created. Currently, BiGym provides 46 high-quality assets that could be reused to facilitate the creation of new environments. In addition to rigid object models, BiGym offers a set of articulated models, such as a dishwasher and customisable kitchen modules.

**API.** The interface for the benchmark follows the standard Gymnasium APIs [44] for training IL and RL agents. A typical workflow of RL agents training is demonstrated in Figure 4.

### 3.3 BiGym Human Demonstration Datasets

One of the key design choices in BiGym is to provide a fixed number of human-collected demonstrations for each task. This allows BiGym benchmark to better reflect the challenges of real-world robot learning, which involves dealing with noisy, multi-modal demonstrations in contrast to synthetic demonstrations generated by motion planners [18]. We describe BiGym's human demonstration dataset collection and management system as below and more details are in the appendix.

**Demo Collection Pipeline.** We use VR[2] to collect demonstrations by virtually tele-operating the H1 robot in simulation (using the 6 DoF poses of the headset and controllers). The headset pose controls the position and orientation of the H1 body, and the 6 DoF poses of the controllers are used to operate the arms. We solve the inverse kinematics for each arm and then reorient the grippers to match the orientation of the respective controller.

**Down-Sampling Demonstrations.** The control frequency of demonstrations can significantly affect the horizon length of tasks and the ability to capture fine grained control, both of which can greatly affect task success [45]. To provide users with the flexibility in selecting action frequencies, we capture demonstrations at the frequency of physics calculation (500 Hz) and provide the functionality to down-sample the demonstrations to a desired frequency (20-500 Hz).

**Demonstration Management.** To minimise the use of storage for saving demonstrations, we save *lightweight* demonstrations that only contain control signals (actions). We then provide a tool that enables users to pull such lightweight demonstrations and replay them to obtain full demonstrations with user-specified observations such as RGB or depth images. These demonstrations are cached in users' local storage so that users do not have to re-download or replay the same demonstrations again.

**Tools.** BiGym provides two tools for demonstration management. The `demo_player` tool provides various demonstration-related functionalities such as downloading, deleting, verifying, replaying at different frequencies, converting, and re-recording demonstrations. The `demo_recorder` enables users to easily collect data by streamlining the process of recording demonstrations with VR.

### 3.4 BiGym Tasks

BiGym provides 40 high-quality tasks with human-collected demonstrations to support the research on household mobile bi-manual manipulation. We describe the tasks and their configurations below.

**Reach Target Tasks.** Different from the standard reach target tasks in single-arm settings, we consider three variants of reach target tasks:

(1) `reach_target_single`: The robot must use a specified wrist to reach a coloured target.
(2) `reach_target_multi_modal`: The robot must reach the target with either left or right wrist. This induces a multi-modal demonstration distribution of reaching the target with different hands. We expect the policy to understand this multi-modality during training.
(3) `reach_target_dual`: In this task, the robot must reach two targets, one with each arm. The success criteria require both wrists to be aligned with corresponding targets. Once this criteria is met, the targets are highlighted to provide visual feedback.

**Table-Top Manipulation Tasks.** We then consider table-top manipulation which requires the robot to interact with rigid-body objects, e.g., plates and cups. The challenge here is that the robot should be

---

[2]Valve Index VR headset, two controllers and two base stations

able to identify remote objects and perform manipulation tasks that require moving the base together with the arms. We introduce the following tasks:

(4) `stack_blocks`: Move blocks across the table, and stack them in the target area.
(5) `move_plate`: Move the plate between two draining racks.
(6) `move_two_plates`: Move two plates simultaneously from one draining rack to the other.
(7) `flip_cup`: Flip the cup, initially positioned upside down on the table, to an upright position.
(8) `flip_cutlery`: Take the cutlery from the static holder, flip it, and place it back into the holder.

In addition to simple single-object manipulation tasks, we introduce complex manipulation tasks that require interaction with articulated objects, where it is crucial to understand the object's kinematics to perform constrained motion planning. The scenarios include: *Dishwasher* and *Kitchen Counter* tasks.

**Dishwasher Tasks.** We consider a set of tasks which require interactions with the articulated dishwasher, ranging from sliding trays to long-horizon unloading tasks:

(9) `dishwasher_open`: Open the dishwasher door and pull out all trays.
(10) `dishwasher_close`: Push back all trays and close the door of the dishwasher.
(11) `dishwasher_open_trays`: Pull out the dishwasher's trays with the door initially open.
(12) `dishwasher_close_trays`: Push the dishwasher's trays back with the door initially open.
(13) `dishwasher_load_plates`: Move plates from the rack to the lower tray of the dishwasher.
(14) `dishwasher_load_cups`: Move cups from the table to the upper tray of the dishwasher.
(15) `dishwasher_load_cutlery`: Move cutlery from the table holder to the dishwasher's cutlery basket. At the beginning of the episode, the dishwasher is open, with the lower tray pulled out.
(16) `dishwasher_unload_plates`: Move plates from the tray of the dishwasher to a table rack.
(17) `dishwasher_unload_cups`: Move cups from the upper tray of the dishwasher to the table.
(18) `dishwasher_unload_cutlery`: Move cutlery from the cutlery basket to a tray on the table.
(19) `dishwasher_unload_plate_long`: A full task of unloading a plate: picking up the plate from dishwasher, placing this plate into the rack located in the closed wall cabinet, and closing the dishwasher and cupboard.
(20) `dishwasher_unload_cup_long`: A full task of unloading a cup: picking up the cup, placing it inside the closed wall cabinet, and closing the dishwasher and cupboard.
(21) `dishwasher_unload_cutlery_long`: A full task of unloading a cutlery: picking up a cutlery, placing it into the cutlery tray inside the closed drawer, and closing the dishwasher and drawer.

**Kitchen Counter Tasks.** In addition, BiGym considers a more complex kitchen counter scenario with multiple challenging articulated objects, e.g., the cupboard, the drawer, etc. Similar to the dishwasher tasks, we provide a range of short and long-horizon tasks as below:

(22) `drawer_top_open`: Open the top drawer of the kitchen cabinet.
(23) `drawer_top_close`: Close the top drawer of the kitchen cabinet.
(24) `drawers_open_all`: Open all sliding drawers of the kitchen cabinet.
(25) `drawers_close_all`: Close all sliding drawers of the kitchen cabinet.
(26) `wall_cupboard_open`: Open doors of the wall cabinet.
(27) `wall_cupboard_close`: Close doors of the wall cabinet.
(28) `cupboards_open_all`: Open all drawers and doors of the kitchen set.
(29) `cupboards_close_all`: Close all drawers and doors of the kitchen set.
(30) `take_cups`: Take two cups out from the closed wall cabinet and put them on the table.
(31) `put_cups`: Pick up cups from the table and put them into the closed wall cabinet.
(32) `pick_box`: Pick up a large box from the floor and place it on the counter.
(33) `store_box`: Move a large box from the counter to the shelf in the cabinet below.
(34) `saucepan_to_hob`: Take the saucepan from the closed cabinet and place it on the hob.
(35) `store_kitchenware`: Take all items from the hob and place them in the cabinet below.
(36) `sandwich_toast`: Use the spatula to put the sandwich on the frying pan.
(37) `sandwich_flip`: Flip the sandwich in the frying pan using the spatula.
(38) `sandwich_remove`: Take the sandwich out of the frying pan.
(39) `store_groceries_lower`: Place a random set of groceries in the cabinets below the counter.

(40) `store_groceries_upper`: Place a random set of groceries in cabinets and shelves on the wall.

**Reward Functions.** We provide sparse rewards for all the tasks based on success detector: a reward of 1 is given for reaching the successful criteria and 0 otherwise. Details are available in Appendix B.

## 4 Experiments

The contribution of the paper is BiGym. However, in this section, we aim to validate that current algorithms can attain some degree of performance on all BiGym tasks, even if minimal. To this end, we conduct experiments with both state-of-the-art IL and demo-driven RL algorithms. Specifically, we focus on the following seven general robot learning algorithms:

**IL Algorithms.** We aim to investigate how different policy representations contribute to the final performance of the algorithms on BiGym, which provides highly noisy and multi-modal demonstrations. In pursuit of this goal, we consider the following algorithms: standard *Behaviour Cloning (BC)*, *Action Chunking Transformers (ACT)* [9] which trains a transformer model [46] to predict a sequence of actions, and *Diffusion Policies [8]* which trains a diffusion model to approximate the expert action distribution. In particular, we do not benchmark against the popular 3D next-best pose agents [10, 11, 16, 17, 19, 20, 21] since they reply on heuristic-based key-frame extraction methods which only apply to single fixed arms [17]; thus, they are not currently applicable to the mobile bi-manual manipulation morphology.

**RL Algorithms.** We mainly consider demo-driven RL algorithms which support training with expert demonstrations. Specifically, we focus on off-policy algorithms and offline RL algorithms that have demonstrated good capabilities in online settings. We consider the following algorithms: *DrQV2* [47], *Advantage Weighted Actor-Critic (AWAC)* [48], *Implicit Q-Learning (IQL)* [49], and *Coarse-to-fine Deep Q-Network (CQN)* [50]. We note that BiGym tasks can be extremely challenging for RL algorithms due to their sparse reward, partial observations, and complex dynamics. To provide a reference for future studies, we provide the results of all the methods as-is with the common set of hyperparameters, instead of tuning their performance for individual BiGym tasks.

We provide experimental results and discussions in Appendix C.

## 5 Discussions

**Limitations.** Currently, BiGym has the following main limitations. Firstly, although BiGym has posed a series of challenging tasks of whole-body mobile bi-manual manipulation, most tasks are still at the skill level, requiring less complex long-horizon task and motion planning, except for the "`_long`" tasks, e.g, `unload_cups_long`. To fully benchmark the capability of algorithms in an ultimate household environment, such tasks are necessary. In addition, BiGym only supports the Unitree H1 robot at the moment. More different embodiments will be included for future tasks.

**Opportunities and Future Works.** BiGym presents various future research opportunities, including but not limited to: (1) exploring better network architectures for approximating multi-modal noisy human demonstrations; (2) studying better belief estimation mechanisms for the POMDP in the mobile manipulation context; (3) investigating better collaboration modes between arms on mobile platforms; (4) whole-body motion planning which improve the efficiency and performance of mobile agents while navigating in cluttered environments.

**Conclusion.** We introduce BiGym, a new and challenging benchmark for demo-driven mobile bi-manual manipulation. BiGym covers 40 challenging tasks of mobile bi-manual manipulation, ranging from simple target reaching to complex dishwasher manipulation tasks. Built upon the humanoid embodiment of Unitree H1 robot, BiGym allows users to flexibly customise the action modes: the whole-body mode and the bi-manual mode. Furthermore, we provide multi-modal and noisy human-collected demonstrations for all BiGym tasks, exhibiting realistic trajectories compared to synthetic ones generated by motion planners. In our experiments, we validate the usability of BiGym by benchmarking state-of-the-art IL and RL algorithms.

## Acknowledgements

Big thanks to the members of the Dyson Robot Learning Lab for discussions and infrastructure help: Iain Haughton, Richie Lo, Sumit Patidar, Sridhar Sola, Mohit Shridhar, Eugene Teoh, Jafar Uruc, and Vitalis Vosylius.

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

# A  Additional Simulation Details

In this section, we provide additional details about BiGym.

**Observation Spaces.** For image observations, we allow users to specify the resolution of the images, where the default resolution is 84×84. Higher resolution may allow learning better policies, but we find the default value works across tasks. In the *whole-body* mode, the proprioception state $s_{\mathrm{proprio}}^{\mathrm{fb}} \in \mathbb{R}^{76} = \{s_{\mathrm{qpos}}^{\mathrm{rb}}, s_{\mathrm{qvel}}^{\mathrm{rb}}, s_{\mathrm{grip}}\}$, where $s_{\mathrm{qpos}}^{\mathrm{rb}} \in \mathbb{R}^{37}$ is the joint angle positions of the robot, $s_{\mathrm{qvel}}^{\mathrm{rb}} \in \mathbb{R}^{37}$ is the corresponding velocities, and $s_{\mathrm{grip}} \in \mathbb{R}^2$ is the gripper opening amount of both grippers. On the contrary, the *bi-manual* mode greatly simplifies the locomotion by replacing the lower-body control with a predefined controller, i.e., a floating base. This reduces the dimension of $s_{\mathrm{qpos}}^{\mathrm{rb}}$ and $s_{\mathrm{qvel}}^{\mathrm{rb}}$ to $s_{\mathrm{qpos}}^{\mathrm{bm}} \in \mathbb{R}^{29}$ and $s_{\mathrm{qvel}}^{\mathrm{bm}} \in \mathbb{R}^{29}$. Furthermore, an additional state $s_{\mathrm{base}} = (x, y, z, \theta) \in \mathbb{R}^4$ is included in $s_{\mathrm{proprio}}$ to indicate the position and orientation of the floating base. As a result, $s_{\mathrm{proprio}}^{\mathrm{bm}} \in \mathbb{R}^{64} = \{s_{\mathrm{qpos}}^{\mathrm{bm}}, s_{\mathrm{qvel}}^{\mathrm{bm}}, s_{\mathrm{base}}, s_{\mathrm{grip}}\}$

**Action Spaces.** In the *whole-body* mode where the agent has the full control over the body, an action space $\mathcal{A}_{\mathrm{wb}} \in \mathbb{R}^{23}$ is defined as $\mathcal{A}_{\mathrm{wb}} = \{\mathcal{A}_{\mathrm{arms}}, \mathcal{A}_{\mathrm{legs}}, \mathcal{A}_{\mathrm{torso}}, \mathcal{A}_{\mathrm{grip}}\}$, where $\mathcal{A}_{\mathrm{arms}} \in \mathbb{R}^{10}$ controls both arms, $\mathcal{A}_{\mathrm{legs}} \in \mathbb{R}^{10}$ controls the legs, $\mathcal{A}_{\mathrm{torso}} \in \mathbb{R}^1$ controls the main torso joints, and $\mathcal{A}_{\mathrm{grip}} \in \mathbb{R}^2$ controls the opening amount of grippers. In *bi-manual* mode, the user controls the floating base instead of the leg joints. Therefore the action space becomes $\mathcal{A}_{\mathrm{bm}} \in \mathbb{R}^{16} = \{\mathcal{A}_{\mathrm{arms}}, \mathcal{A}_{\mathrm{base}}, \mathcal{A}_{\mathrm{grip}}\}$, with $\mathcal{A}_{\mathrm{base}} \in \mathbb{R}^4$ controlling the delta actions $(\delta x, \delta y, \delta z, \delta\theta)$ of the base.

**Simulation Performance.** We present the simulation speed in Figure 5. The benchmark was done on a headless server of NVIDIA L4 GPU and Intel Xeon Gold 6438Y+ CPU, in a single process. Benefiting from the highly optimised MoJoCo engine, BiGym runs at around 400FPS to 1400FPS depending on the number of cameras. The performance could be further improved by using parallel environments or MuJoCo XLA, which speeds up the execution with XLA just-in-time compilation.

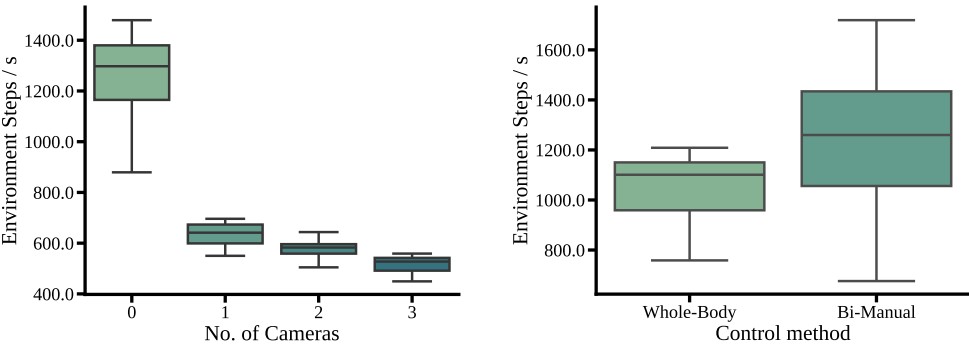

(a) Environment Speed with Different # Cameras    (b) Environment Speed of Different Action Modes

Figure 5: The environment run speed of BiGym with (a) different number of cameras and (b) different action modes. In (a), we use the bi-manual control method for measuring the performance.

# B  Details of Task Success Detectors

In this section, we detail the definitions of all task success detectors.

**Reach Target Tasks.**

(1) `reach_target_single`: The distance from the robot left wrist to the target is smaller than a tolerance value. The default tolerance value is 0.1.
(2) `reach_target_multi_modal`: The distance from either the robot left wrist or the right wrist is smaller than a tolerance value. The default tolerance value is 0.1.
(3) `reach_target_dual`: The distance from the left wrist and the right wrist to their corresponding goals are smaller than a tolerance value. The default tolerance value is 0.1.

**Table-Top Manipulation Tasks.**

(4) `stack_blocks`: The three blocks are stacked on each other, i.e. in collision with each other, in a target region on the table.

(5) `move_plate`: The following conditions must be met: (a) the orientation of the plate is upright, (b) the plate is not colliding with the table, (c) the plate is colliding with the rack and (d) the robot has released the plate from its gripper.

(6) `move_two_plates`: Transfer two plates to the target rack and meet all conditions similar to the `move_plate` task.

(7) `flip_cup`: The following criteria must be met: (a) The cup is in collision with the counter. (b) The orientation of the cup is upright. (c) The robot has released the cup from its gripper.

(8) `flip_cutlery`: Similar to the `flip_cup` task, but cutlery is used instead.

**Dishwasher Tasks.**

(9) `dishwasher_open`: The joint angles of the dishwasher door and both trays are close to 1 with a tolerance value. The default value is 0.1.

(10) `dishwasher_close`: The joint angles of the dishwasher door and both trays are close to 0 with a tolerance value. The default value is 0.1.

(11) `dishwasher_open_trays`: The joint angles of dishwasher trays are close to 1 with a tolerance value. The default value is 0.1.

(12) `dishwasher_close_trays`: The joint angles of dishwasher trays are close to 0 with a tolerance value. The default value is 0.1.

(13) `dishwasher_load_plates`: All plates are in collision with the bottom tray of the dishwasher and the robot has released the plates from it's grippers.

(14) `dishwasher_load_cups`: All cups are in collision with the middle tray of the dishwasher and the robot has released the cup from its gripper.

(15) `dishwasher_load_cutlery`: All cutlery are in collision with the dishwasher cutlery basket and the robot has released the cutlery from its gripper.

(16) `dishwasher_unload_plates`: All plates are moved from the bottom tray of the dishwasher to the drainer on the table, and placed onto the rack positioned on the counter-top.

(17) `dishwasher_unload_cups`: All cups are moved from the middle tray of the dishwasher to the cabinet and in collision with the cabinet counter. Additionally, all cups are released from the robot gripper.

(18) `dishwasher_unload_cutlery`: All cutlery are moved from the dishwasher basket to the tray and are in collision with the tray.

(19) `dishwasher_unload_plate_long`: All conditions of `dishwasher_close` and `dishwasher_unload_plates` must be met. Additionally, all plates are placed inside the wall cabinet. Finally, all joint angles of the wall cabinet doors are close to 0 with a tolerance. The default value is 0.1.

(20) `dishwasher_unload_cup_long`: Similar to `dishwasher_unload_plate_long` but with cups.

(21) `dishwasher_unload_cutlery_long`: Similar to `dishwasher_unload_cutlery_long` but with cutlery and instead of the cabinet, the cutlery must be placed in a closed drawer.

**Kitchen Counter Tasks.**

(22) `drawer_top_open`: The joint angle of the top drawer is close to 1 with a tolerance value. The default value is 0.1.

(23) `drawer_top_close`: The joint angle of the top drawer is close to 0 with a tolerance value. The default value is 0.1.

(24) `drawers_open_all`: The joint angles of all drawers are close to 1 with a tolerance value. The default value is 0.1.

(25) `drawers_close_all`: The joint angles of all drawers are close to 0 with a tolerance value. The default value is 0.1.

(26) `wall_cupboard_open`: The joint angle of two doors of the wall cupboard is close to 1 with a tolerance value. The default value is 0.1.

(27) `wall_cupboard_close`: The joint angle of two doors of the wall cupboard is close to 0 with a tolerance value. The default value is 0.1.

(28) `cupboards_open_all`: The joint angles of the two doors and all drawers of the kitchen set are close to 1 with a tolerance value. The default value is 0.1.

(29) `cupboards_close_all`: The joint angles of the two doors and all drawers of the kitchen set are close to 0 with a tolerance value. The default value is 0.1.

(30) `take_cups`: All cups are in collision with the counter on the table and the robot has released the cups from its gripper.

(31) `put_cups`: All cups are in collision with the cupboard shelf and the robot has released the cups from its gripper.

(32) `pick_box`: The box is in collision with the counter and the robot has released the box from its grippers.

(33) `store_box`: The box is in collision with the shelf and the robot has released the box from its grippers.

(34) `saucepan_to_hob`: The saucepan is in collision with the hob and the robot has released the saucepan from its grippers.

(35) `store_kitchenware`: Both the saucepan and the pan are in collision with the shelf, and the robot has released the objects from its grippers.

(36) `sandwich_toast`: All the following conditions must be met: (a) The sandwich is in collision with the pan. (b) The orientation of the sandwich is either up or down. (c) The pan is in collision with the hob.

(37) `sandwich_flip`: Similar to `sandwich_toast`. In addition the sandwich orientation must be flipped.

(38) `sandwich_remove`: All the following conditions must be met: (a) The sandwich is in collision with the board. (b) The orientation of the sandwich is either up or down.

(39) `store_groceries_lower`: All items are in collision with the shelf of the cabinet below the counter. Additionally, all items are released from the robot gripper.

(40) `store_groceries_upper`: All items are in collision with the shelf of the cabinet on the wall. Additionally, all items are released from the robot gripper.

## C  Experiments

### C.1  Implementation Details

We implemented all algorithms using PyTorch [51].

**ACT**. Following the official implementation[3], we train a ResNet-18 encoder [52] to extract visual features and a transformer model to predict a sequence of actions. Inputs to the transformer model are multi-view image features and proprioceptive features from a conditional variational autoencoder (CVAE) [53]. During execution, we use receding horizon control for all tasks by training the policy to output an action sequence of length 16 and executing only the first step in the sequence. Following the official implementation, we enable *temporal ensembling* to improve the smoothness of the policy.

**Diffusion Policy**. Our implementation of Diffusion Policy closely follows the official release[4]. To be consistent with ACT, we use ResNet-18 as vision encoders for all camera observations. As discussed in Chi et al. [8], the Diffusion Policy is susceptible to the choice of backbones and their parameters: the UNet-1D backbone might outperform the causal transformer backbone in certain tasks and vice versa. Thus, we benchmark both the UNet-1D backbone and the causal Transformer backbone, and report the highest achieved performance between them in our main results. In addition, for all Diffusion Policy variants, we use action sequence length of 16 and execution length of 1, which we find to achieve strong performance in general. Following ACT, we also enable *temporal ensembling* for Diffusion Policies, which we find to be crucial for stabilising the inference.

---

[3]https://github.com/tonyzhaozh/aloha
[4]https://github.com/real-stanford/diffusion_policy

**Other Baselines**. For BC and demo-driven RL baselines, we adopt the same network architectures which consist of an CNN-based image encoder and a fully-connected output head. The image encoder encodes each camera image with 3 layers of CNNs, each has kernel size 3 and 32 channels. In between the layers, we use SiLU activation function [54] and layer normalisation [55]. We flatten the CNN features and concatenate with the proprioception states to form the final observation feature vector. The head has 2 fully connected layers of dimension 512, and bottlenecks the output to dimension 64. After the bottleneck layer, we normalise the output with layer normalisation followed by tanh activation.

**Training Details.** We use a frame stack of 4, Adam optimiser [56] with a learning rate of 0.0001, and batch size of 256 for all IL and demo-driven RL algorithms. In addition, specifically for all RL algorithms, we follow AW-Opt [57] and keep the demonstration ratio for each batch to be 50% by using a separate demonstration replay buffer. This helps the exploration of the agent during sparse reward settings. We run 150K training steps for IL algorithms and 100K steps for demo-driven RL methods. We observe that all algorithms converge after 100K steps and longer training does not give additional performance boost. All results are averaged over the last 3 checkpoints.

## C.2 Results and Discussions

In Table 2, we provide the performance of IL and demo-driven RL methods on 40 BiGym tasks. Overall, we observe that BiGym tasks are challenging and pose a variety of unique and interesting challenges for future researches. We outline our observations as below:

**The mobile manipulation of articulated or rigid-body objects is challenging for the state-of-the-art algorithms.** BiGym has presented a series of tasks that involve interactions with articulated or rigid-body objects, which typically require high-precision manipulation, e.g., `move_two_plates`, `cupboards_open_all`, and `stack_blocks`. When coupled with the mobile base, these tasks become more challenging because (i) accurately measuring the grasping poses while moving is hard, and (ii) correctly estimating the posterior distribution of the states given partial history information of a POMDP is difficult. For instance, while we observe that ACT and Diffusion Policy achieve the overall best performance across all tasks, they still struggle in the seemingly simple tasks, e.g., `stack_blocks`, which requires the agent to pick 3 cubes, and stack them to a target region located on the other side of the table. We believe a more robust system with stronger memory mechanisms to track the "beliefs", i.e., estimating the posterior distributions of the states, is necessary to solve such challenging BiGym tasks.

**The long-horizon tasks in BiGym requires both task and motion planning of the agent.** BiGym introduces a series of long-horizon tasks, e.g., `dishwasher_unload_cups_long` and `put_cups`. All algorithms fail on these tasks. Intuitively, these tasks are composed of multiple sub-tasks, and the difficulty level of achieving these long-horizon tasks grows exponentially at the same time. As model-free agents, our baselines are not capable of performing task-level reasoning. Hierarchical methods [12] could work as a better policy representation for these tasks. We leave it for future study.

**The complex policy space of BiGym requires carefully designed agent architectures.** We observe that almost on all tasks, ACT and Diffusion Policy achieves superior performance to BC and demo-driven RL baselines. We hypothesize this is because both ACT and Diffusion Policy utilise powerful policy classes based on generative representation learning, i.e., CVAE and Diffusion models, and they also use expressive network architecture such as transformers or UNets. In contrast, BC and all demo-driven RL approaches use simple CNN + MLP architectures. It is likely that these weaker architectures struggle to deal with the complex multi-modal noisy demonstrations introduced in BiGym. We believe this can motivate future research on finding appropriate policy representations for mobile bi-manual manipulation.

**Demo-driven RL approaches struggle with the complex task space and sparse reward in BiGym.** We observe that demo-driven RL algorithms fail on most of the BiGym tasks. For instance, CQN [50], which exhibits strong performance on fixed single-arm demo-driven RL setups, fails to solve most of the BiGym tasks. It is notable that all RL algorithms only achieve non-zero success rates on simple

Table 2: Success rates (%) of IL and demo-driven RL algorithms on 40 BiGym tasks, evaluated on 50 episodes. We report the results aggregated over the last three checkpoints.

| Task | IL Algorithms | | | RL Algorithms | | | |
|---|---|---|---|---|---|---|---|
| | BC | ACT | DiffPolicy | DrQV2 | AWAC | IQL | CQN |
| reach_target_single | 66.0±0.0 | **100.0±0.0** | 61.3±5.8 | **100.0±0.0** | 94.0±2.0 | 82.0±5.3 | 92.7±1.2 |
| reach_target_multi_modal | 75.3±2.3 | **98.7±1.2** | 63.3±3.1 | **100.0±0.0** | **100.0±0.0** | 53.3±8.1 | 69.3±2.3 |
| reach_target_dual | 23.3±2.3 | **90.7±1.2** | 19.3±3.1 | 24.0±2.0 | 77.3±6.1 | 48.7±20.2 | 40.0±10.6 |
| stack_blocks | 0.0±0.0 | 0.0±0.0 | 0.0±0.0 | 0.0±0.0 | 0.0±0.0 | 0.0±0.0 | 0.0±0.0 |
| move_plate | 2.7±1.2 | **30.0±3.5** | 20.0±2.0 | 0.0±0.0 | 0.0±0.0 | 0.0±0.0 | 0.7±1.2 |
| move_two_plates | 7.3±2.3 | 11.3±7.0 | **12.0±4.0** | 0.0±0.0 | 0.0±0.0 | 0.0±0.0 | 0.0±0.0 |
| flip_cup | 0.0±0.0 | **21.3±1.2** | 6.0±2.0 | 0.0±0.0 | 0.0±0.0 | 1.3±1.2 | 0.0±0.0 |
| flip_cutlery | 0.7±1.2 | **22.0±2.0** | 1.3±1.2 | 0.0±0.0 | 0.7±1.2 | 1.3±1.2 | 1.3±1.2 |
| dishwasher_open | 6.0±5.3 | **72.0±45.0** | 4.0±4.0 | 0.0±0.0 | 0.0±0.0 | 0.0±0.0 | 0.0±0.0 |
| dishwasher_close | 84.7±20.0 | **100.0±0.0** | 99.3±1.2 | 0.0±0.0 | 0.0±0.0 | 0.0±0.0 | 0.0±0.0 |
| dishwasher_open_trays | 16.7±5.8 | **100.0±0.0** | 0.0±0.0 | 0.0±0.0 | 0.0±0.0 | 0.0±0.0 | 0.0±0.0 |
| dishwasher_close_trays | 0.0±0.0 | **100.0±0.0** | 52.0±18.3 | 2.0±2.0 | 0.0±0.0 | 0.0±0.0 | 0.0±0.0 |
| dishwasher_load_plates | 0.0±0.0 | **34.0±8.7** | 0.0±0.0 | 0.0±0.0 | 0.0±0.0 | 0.0±0.0 | 0.0±0.0 |
| dishwasher_load_cups | 0.0±0.0 | **46.0±0.0** | 8.7±5.0 | 0.0±0.0 | 0.0±0.0 | 0.0±0.0 | 0.0±0.0 |
| dishwasher_load_cutlery | 8.7±2.3 | **42.0±8.7** | 3.3±2.3 | 0.0±0.0 | 0.0±0.0 | 0.0±0.0 | 0.0±0.0 |
| dishwasher_unload_plates | 5.3±1.2 | 2.0±3.5 | 0.0±0.0 | 0.0±0.0 | 0.0±0.0 | 0.0±0.0 | 0.0±0.0 |
| dishwasher_unload_cups | 9.3±4.2 | **15.3±10.1** | 0.7±1.2 | 0.7±1.2 | 0.7±1.2 | 0.0±0.0 | 0.0±0.0 |
| dishwasher_unload_cutlery | 3.3±2.3 | **18.0±3.5** | 1.3±1.2 | 0.0±0.0 | 0.0±0.0 | 0.0±0.0 | 0.0±0.0 |
| dishwasher_unload_plates_long | 0.0±0.0 | 0.7±1.2 | 0.0±0.0 | 0.0±0.0 | 0.0±0.0 | 0.0±0.0 | 0.0±0.0 |
| dishwasher_unload_cups_long | 0.0±0.0 | 0.0±0.0 | 0.0±0.0 | 0.0±0.0 | 0.0±0.0 | 0.0±0.0 | 0.0±0.0 |
| dishwasher_unload_cutlery_long | 1.3±2.3 | **14.7±8.3** | 5.3±5.8 | 0.0±0.0 | 0.0±0.0 | 0.0±0.0 | 0.0±0.0 |
| drawer_top_open | 9.3±16.2 | **100.0±0.0** | 3.3±3.1 | 0.0±0.0 | 8.7±15.0 | 2.0±2.0 | 0.0±0.0 |
| drawer_top_close | **100.0±0.0** | **100.0±0.0** | **100.0±0.0** | **100.0±0.0** | **100.0±0.0** | **100.0±0.0** | **100.0±0.0** |
| drawers_open_all | 10.7±10.1 | **100.0±0.0** | 16.7±8.3 | 0.0±0.0 | 0.0±0.0 | 0.0±0.0 | 0.0±0.0 |
| drawers_close_all | 0.0±0.0 | **100.0±0.0** | 27.3±18.6 | **100.0±0.0** | **100.0±0.0** | **100.0±0.0** | 44.0±10.4 |
| wall_cupboard_open | 22.0±31.2 | 97.3±1.2 | **100.0±0.0** | 27.3±5.0 | 12.0±17.3 | 9.3±2.3 | 0.0±0.0 |
| wall_cupboard_close | **100.0±0.0** | **100.0±0.0** | **100.0±0.0** | **100.0±0.0** | 26.0±41.6 | 97.3±4.6 | 70.0±2.0 |
| cupboards_open_all | 5.3±4.2 | **17.3±21.4** | 0.0±0.0 | 0.0±0.0 | 0.0±0.0 | 0.0±0.0 | 0.0±0.0 |
| cupboards_close_all | **63.3±7.0** | 0.7±1.2 | 1.3±2.3 | 0.0±0.0 | 0.0±0.0 | 0.0±0.0 | 0.0±0.0 |
| take_cups | 0.0±0.0 | **26.0±2.0** | 5.3±2.3 | 0.0±0.0 | 0.0±0.0 | 0.0±0.0 | 0.0±0.0 |
| put_cups | 3.3±2.3 | **30.0±7.2** | 0.7±1.2 | 0.0±0.0 | 0.0±0.0 | 0.0±0.0 | 0.0±0.0 |
| pick_box | 20.7±1.2 | **40.7±1.2** | 22.0±0.0 | 0.0±0.0 | 0.0±0.0 | 0.0±0.0 | 0.0±0.0 |
| store_box | 8.7±3.1 | **13.3±3.1** | 0.0±0.0 | 0.0±0.0 | 0.0±0.0 | 0.0±0.0 | 0.0±0.0 |
| saucepan_to_hob | 21.3±4.6 | **88.0±2.0** | 34.7±3.1 | 0.0±0.0 | 0.0±0.0 | 0.0±0.0 | 0.0±0.0 |
| store_kitchenware | 0.0±0.0 | 2.7±2.3 | 0.7±1.2 | 0.0±0.0 | 0.0±0.0 | 0.0±0.0 | 0.0±0.0 |
| sandwich_toast | 5.3±1.2 | **30.7±6.1** | 10.0±0.0 | 0.0±0.0 | 0.0±0.0 | 0.0±0.0 | 0.0±0.0 |
| sandwich_flip | 0.0±0.0 | **32.0±2.0** | 4.7±1.2 | 0.7±1.2 | 0.0±0.0 | 0.0±0.0 | 0.0±0.0 |
| sandwich_remove | 40.7±8.3 | **55.3±7.0** | 48.0±0.0 | 0.7±1.2 | 0.0±0.0 | 0.0±0.0 | 0.0±0.0 |
| store_groceries_lower | 0.0±0.0 | 0.0±0.0 | 0.0±0.0 | 0.0±0.0 | 0.0±0.0 | 0.0±0.0 | 0.0±0.0 |
| store_groceries_upper | 0.0±0.0 | 0.0±0.0 | 0.0±0.0 | 0.0±0.0 | 0.0±0.0 | 0.0±0.0 | 0.0±0.0 |
| Average | 18.0±1.1 | **46.3±1.4** | 20.8±0.8 | **13.9±0.2** | 13.0±1.2 | 12.4±0.0 | 10.5±0.4 |

tasks with little interaction with the objects, e.g., reach_target_single and top_drawer_close, and completely fail to solve all the other tasks. We hypothesise this is because (i) the presence of mobile base makes it more difficult for agents to explore meaningful state. e.g. an erroneous base turning action can easily cause robot to lose view of the objects. and (ii) RL agents struggle to learn value-functions on long-horizon BiGym tasks with sparse reward.

