# OpenReview forum: "BiGym: A Demo-Driven Mobile Bi-Manual Manipulation Benchmark"
_robot-learning.org/CoRL/2024/Conference — CoRL 2024_

### Official Review · Reviewer_bdfR · 2024-07-20
**The paper is structured and easy-to-follow, however, there are a few concerns need to be addressed.**

**Originality:** 3
**Technical Quality:** 4
**Clarity Of Presentation:** 5
**Potential Impact:** 3
**Recommendation:** 3
**Confidence:** 4

**Review:**

This paper introduces BiGym, a new benchmark and learning environment for mobile bi-manual robotic manipulation.

The paper is easy-to-follow, and I appreciate a lot on the literature review as it distinct itself from existing works.

The key contributions and strengths of BiGym include:
- A diverse set of 40 tasks in home environments, ranging from simple target reaching to complex kitchen cleaning scenarios. This provides a good breadth of challenges.
- Human-collected demonstrations for each task, which capture more realistic and multi-modal trajectories compared to synthetic planner-generated demos used in some previous benchmarks.
- Support for multiple observation types including proprioceptive data and visual inputs from 3 camera views. This flexibility is valuable.
- Benchmarking of state-of-the-art imitation learning and demo-driven reinforcement learning algorithms to validate the usability of the environment.
- A flexible framework allowing users to switch between whole-body control and simplified bi-manual control modes.

Some comments:
- The robot's bipedal movement seem unrealistic from the simulation? I therefore don't think the data is useful for whole body mobile manipulation
- I would like to see a bit of reasoning for the selected tasks.
- I also didn't see a limitation section?

**Quality Of The Limitations Section:**

1

**Questions For Rebuttal:**

The bipedal movement seems very unrealistic, how are you going to resolve this issue?
The reason of selecting the tasks is unclear to me.
Elaboration on limitation?

**Robotics Focus:**

2

**Summary Of Paper:**

The paper presents a benchmark data set for (mobile) bimanual manipulation tasks. The data is collected from human demo with a comprehensive bimanual tasks.

**Summary Of Recommendation:**

I think this is a good paper for CoRL if my concerns are addressed.

---

### Official Review · Reviewer_JNyU · 2024-07-20

**Originality:** 4
**Technical Quality:** 3
**Clarity Of Presentation:** 4
**Potential Impact:** 4
**Recommendation:** 3
**Confidence:** 4

**Review:**

Strengths:
+ Timely research. A benchmark suite on humanoid bimanual manipulation with demonstration data is of great interest to the robot learning community.
+ Code will be released upon publication.
+ The paper presents baseline results for most popular BC and offline RL methods, which will be useful reference points for future research.


Weaknesses:
- While-body control mode is claimed as a main contribution of the work. However, I can’t seem to find any details about the full-body controller while the robot is being teleoperated. How is it implemented? From the demonstration video, it seems that the leg and the torso are independently controlled: torso translation and rotation do not seem to correlate with the leg movement. For example, the legs maintain the same gaiting while the robot is crouching, which seems physically impossible. Also the upper body rotation seems to be limited to yaw and has no roll or pitch in the videos.
- If the whole-body control mode is indeed simplified, I’d like to better understand how this affects the realism of the benchmark. For example, most full-body tasks such as lifting heavy items require complex coordination of upper and lower body motion. If the benchmark cannot expose these kinds of challenges, would algorithms tested in simulation reflect their real-world performances?

=== Post rebuttal:

The authors have addressed some of my concerns, e.g., all the videos demonstrated are actually bimanual mode, not whole-body mode. This left me wondering what the whole-body control mode would look like. Regardless, I believe Bigym would be of wide interest to the CoRL community and I therefore have updated my rating to weak accept.

**Quality Of The Limitations Section:**

2

**Questions For Rebuttal:**

See "weaknesses"

**Robotics Focus:**

3

**Summary Of Paper:**

The paper presents a simulated benchmark and demonstration dataset for bimanual manipulation tasks based on the UniTree H1 humanoid robot. The benchmark features a variety of bimanual tasks situated in a kitchen environment. It supports both full-body control and bimanual mode with a floating base. The paper also includes results in comparing popular behavior cloning methods and offline RL methods.

**Summary Of Recommendation:**

I'm willing to change my rating if the authors can adequately address my question regarding full-body control.

---

### Official Review · Reviewer_RTpp · 2024-07-21
**New MuJoCo-based mobile bi-manual manipulation benchmark**

**Originality:** 2
**Technical Quality:** 3
**Clarity Of Presentation:** 5
**Potential Impact:** 3
**Recommendation:** 3
**Confidence:** 4

**Review:**

**Pros**

- Timely support for both learning paradigms - provide collected demo for each task to support benchmarking both RL and imitation learning algorithms.

- Flexible as a benchmark - Supports both locomotion and mobile bi-manual manipulation, and allows switching between whole-body and bi-manual to focus on whole-body locomotion with manipulation, or focus on manipulation specifically.

- Includes relatively extensive benchmarking on current state of the art algorithms in IL and RL on the provided tasks.

- Open-sourced benchmarking will benefit the community and push for better benchmarking for mobile bimanual tasks!

- Provide a longer task horizon and the option of treating the lower-body as floating base controlled by the classical controller, to focus on upper-body bi-manual skills.

- The supplement provides extensive benchmarking experiment results between IL methods (BC, ACT, Diffusion Policy) and RL (DrQV2, AWAC, IQL, CQN) on the provided tasks, and finds that RL generally suffer in performance compared to IL methods.


**Cons**

- Right now, the simulator only supports Unitree H1 humanoid with robotiq parallel gripper, it would be great for future iterations to include dextrous hands and dextrous manipulation tasks that require dexterous hands.

- The tasks included in this benchmark is fairly simple and easy for humanoid/bimanual setup. Future iterations could benefit from including tasks that require more whole-body manipulation, including using the robot arm in addition to robot end effector to complete the tasks.

**Others**

- It would be very helpful for the authors to provide some evaluation experiments on sim2real transfer gap. For example, some of the table-top task setup could be replicated in real world, and some preliminary real-world eval results on the policy trained from simulation would be useful to provide insights on how physically transferable the policy trained from simulation physics to real world.

**Quality Of The Limitations Section:**

2

**Questions For Rebuttal:**

- It would be very helpful for the authors to provide some evaluation experiments on sim2real transfer gap. For example, some of the table-top task setup could be replicated in real world, and some preliminary real-world eval results on the policy trained from simulation would be useful to provide insights on how physically transferable the policy trained from simulation physics to real world.

**Robotics Focus:**

3

**Summary Of Paper:**

The authors proposed BiGym -  a benchmark based on MuJoCo and learning environment for mobile bi-manual manipulation with 40 tasks, with 50 human-collected demos for each task. It provides visual inputs as RBGD with overhead and wrist camera views. The paper provides extensive benchmarking results of current SOTA RL and IL methods.

**Summary Of Recommendation:**

Great system paper with a new benchmark to push for mobile bi-manual manipulation, but more real-world result is needed to distinguish it from concurrent related works.

---

### Official Review · Reviewer_oipX · 2024-07-22
**Extensive mobile bimanual benchmark with 40 skills of ranging difficulty, but baseline evaluation should be included in the main paper and expanded.**

**Originality:** 3
**Technical Quality:** 3
**Clarity Of Presentation:** 4
**Potential Impact:** 3
**Recommendation:** 3
**Confidence:** 3

**Review:**

Strengths:
* The paper topic is pertinent to the research community as mobile manipulation/humanoid hardware has become popular. A benchmark in this space it timely and needed by the research community.
* Generally the paper is well written and easy to read.
* The tasks and design choices are detailed, explained, and illustrated.
* The experiments clearly show a gap on the benchmark for the considered baselines and the discussion of the results is reasonable and insightful.

Weaknesses:
* The experimental results (or some subset of them) should really be in the main paper. There is a lot of repetition currently in the main paper regarding the 40 tasks and the contents of the benchmark. I would cut down on some of the repetition in favor of including the experimental results and discussion.
* The discussion suggests that longer temporal memory could help ACT/diffusion policy do better on the long-horizon tasks, while hierarchical RL methods could help with the sparse reward. Although I do not expect these changes to completely solve the benchmark (nor should they), it would be pertinent to show that these are reasonable directions. Adding some memory to diffusion policy for example and trying out an existing hierarchical RL algorithm seems like a reasonable course of action.
* It would be great to demonstrate that the simulation findings transfer to hardware.
* The limitation section is missing.

* The following is a non-exhaustive list of typos and grammatic errors.
1. Line 127: 'which supports a diverse types of algorithms which use either type of observation' - incorrect grammar and poor wording.
2. Line 255: 'since they reply on'.
3. Line 569: 'The long horizon tasks in BiGym requires'.
4. Line 574: 'We leave [this] for future work.'
5. The references have some consistency issues and need to be proofread for typos. For example, conference acronyms are sometimes included (e.g., [8]) and sometimes not (e.g., [1]). Some conference names were not capitalized (e.g., [1]). When possible, the conference venue should be cited instead of ArXiv. The references should in general be proofread (e.g., capitalization errors in the title of [4, 11], the year is listed twice in [42], etc.).

**Quality Of The Limitations Section:**

1

**Questions For Rebuttal:**

In addition to the weaknesses listed above, I have the following questions for the authors.
1. Is 50 demonstrations per skill enough, particularly for long horizon tasks? How was this number chosen?
2. Is saving action-only demonstrations that can be replayed later to get the observations sufficient? During replaying, are the rollouts of the demonstrations the same or stochastically different?

**Robotics Focus:**

3

**Summary Of Paper:**

The paper presents a mobile bimanual benchmark in simulation with 40 human-demonstrated skills (50 demos each) of ranging difficulty. The skills include reach target, table-top manipulation, dishwasher, and kitchen counter tasks. The skills are benchmarked using 3 imitation learning baselines and 4 RL baselines. The results demonstrate that ACT and diffusion policy imitation learning algorithms perform best but leave a lot of room for improvement, and even completely fail on some tasks that are long-horizon and/or involve mobile navigation and manipulation. The conclusions are we need algorithms with longer temporal memory and for RL potentially hierarchical approaches to deal with the sparse reward on long-horizon tasks.

**Summary Of Recommendation:**

Overall, I think the mobiel bimanual benchmark is a good addition to the research community. My main points of feedback are: include the experiments in the main paper, try out the couple of ideas suggested (temporal memory for imitation learning and hierarchical RL), and demonstrate that the skills from the simulation benchmark transfer in some way to hardware.

---

### Decision · Program_Chairs · 2024-09-04

**Decision:**

Accept

**Comment:**

Strengths:
- Interesting and timely topic of bimanual manipulation, with a rich set of tasks consisting of both bimanual and whole-body manipulation.
- Reviewers agree that the baselines and experiments are insightful, showing gaps in SoTA on the proposed benchmark.

Weaknesses:
- Reviewers raise concerns about the whole-body motion demos and controller and its impact on the benchmark’s realism, especially sim2real.
- The experiments and analysis should be in the main paper, instead of in the appendix. Details about the controller used for collection demonstrations, and a limitation section should also be added to the paper.

Post-rebuttal:

The key points addressed in the rebuttal include:
- Limitation section: Authors have added a limitations section to the paper.
- Experimental details: Authors have described details of the controller and teleoperation.
However, some concerns raised by reviewers around sim2real still remain.